# Focal Liver Lesions in Budd-Chiari Syndrome: Spectrum of Imaging Findings

**DOI:** 10.3390/diagnostics13142346

**Published:** 2023-07-12

**Authors:** Francesco Rizzetto, Davide Rutanni, Luca Alessandro Carbonaro, Angelo Vanzulli

**Affiliations:** 1Department of Radiology, ASST Grande Ospedale Metropolitano Niguarda, Piazza Ospedale Maggiore 3, 20162 Milan, Italy; luca.carbonaro@unimi.it (L.A.C.); angelo.vanzulli@unimi.it (A.V.); 2Postgraduate School of Diagnostic and Interventional Radiology, Università degli Studi di Milano, Via Festa del Perdono 7, 20122 Milan, Italy; davide.rutanni@unimi.it; 3Department of Oncology and Hemato-Oncology, Università degli Studi di Milano, Via Festa del Perdono 7, 20122 Milan, Italy

**Keywords:** Budd-Chiari syndrome, review, liver neoplasms, hepatocellular carcinoma, focal nodular hyperplasia, computed tomography, magnetic resonance imaging, ultrasonography

## Abstract

Budd–Chiari syndrome (BCS) is a rare clinical entity characterized by hepatic venous outflow obstruction, resulting in liver congestion and subsequent chronic parenchymal damage. This condition often leads to the development of focal liver lesions, including benign focal nodular hyperplasia-like regenerative nodules, hepatocellular carcinoma, and perfusion-related pseudo-lesions. Computed tomography, ultrasound, and magnetic resonance are the commonly employed imaging modalities for the follow-up of BCS patients and for the detection and characterization of new-onset lesions. The accurate differentiation between benign and malignant nodules is crucial for optimal patient management and treatment planning. However, it can be challenging due to the variable and overlapping characteristics observed. This review aims to provide a comprehensive overview of the imaging features and differential diagnosis of focal liver lesions in BCS, emphasizing the key findings and discussing the challenges associated with their interpretation, with the purpose of facilitating the subsequent clinical decision-making.

## 1. Introduction

Budd–Chiari syndrome (BCS) is a rare disorder characterized by obstruction of hepatic venous outflow. The most prevalent form, defined as primary, is related to hypercoagulable conditions that lead to the formation of blood clots in the hepatic veins, affecting either the distal branches or the main vessels [1]. In rarer instances, primary BCS is caused by a membranous web-like obstruction in the inferior vena cava, usually of congenital origin, which may extend to the hepatic veins [2]. Conversely, secondary BCS occurs when the hepatic veins are externally compressed by an expansive mass or invaded by neoplastic growths [3].

Regardless of the underlying cause, BCS results in liver congestion with hepatocellular necrosis and atrophy, leading to the development of portal hypertension and parenchymal fibrosis within a few weeks of the venous obstruction. Consequently, liver regeneration mechanisms activate, including responsive hyperarterialization and hypertrophy in regions with preserved venous drainage [4]. These processes are responsible for hepatocellular proliferation, thereby facilitating the emergence of various types of lesions, such as regenerative nodules and hepatocellular carcinoma (HCC). Accurately distinguishing between benign and malignant lesions is crucial due to its significant implications for patient management and treatment decisions. In this regard, computed tomography (CT), magnetic resonance imaging (MRI), and ultrasound (US) serve as the initial non-invasive diagnostic modalities for the detection and characterization of focal liver lesions in BCS.

## 2. Focal Nodular Hyperplasia-like Regenerative Nodules

Focal nodular hyperplasia-like regenerative nodules (FNH-like RNs) are the most common liver lesions in patients with BCS, with a reported prevalence of 36–44% [5,6]. These nodules, documented only in chronic disease, seem to develop because of the combination of impaired liver venous drainage, portal flow deprivation and increased hepatic arterialization. The parenchymal damage due to venous congestion is supposed to induce hepatocellular proliferation in the areas with preserved venous outflow, a process further supported by arterial neoformation and consequent hyperperfusion [7].

Histologically, these nodules are composed of hyperplastic hepatocytes and fibrovascular septa, tend to present scarce-to-absent portal supply, and may display large arteries branching radially from the lesion fibrous center, closely resembling FNH [8,9,10]. However, since FNH typically develops in healthy liver parenchyma [11], the definition of FNH-like RNs was suggested to emphasize their distinct nature [12]. As different nodules in the same liver can show different microscopic findings, and mixed patterns may be observed in the same nodule [13], FNH-like RNs in BCS could even be assumed to express a lesion spectrum between regenerative nodules and FNH. Unusual pathological characteristics include intranodular cholestasis and fatty degeneration [10], whereas hemorrhage and calcification have not been observed [14].

FNH-like RNs can involve all liver segments, with a predilection for the right liver (67–82%) [15,16] probably secondary to the right lobe atrophy and left lobe hypertrophy that characterize chronic stage BCS. To confirm the role of imbalanced vascular supply, a longitudinal study by Flor et al. [15] found that only 1% of FNH-like RNs were located in the caudate lobe. Indeed, the venous outflow of this segment, the only one to directly drain into the inferior vena cava, is usually preserved in BCS [17]. 

FNH-like RNs are most often numerous and variably sized, with a diameter usually ranging from 0.5 to 4 cm [18], although the lesions detectable on CT and MR imaging are only a tiny fraction of all the nodules actually present, with most of them being very small [14]. No specific distribution has been identified, although microscopic nodules show preferential localization in the periportal regions [19,20]. 

During follow-up the nodules can increase in size and/or number, likely contributing with their expansile growth to portal hypertension through compression of the central vein and portal spaces [7,18]. New FNH-like RNs may also appear after transjugular intrahepatic portosystemic shunt (TIPS) positioning, as a consequence of the hemodynamic changes that the procedure induces in the liver parenchyma [21]. However, up to 40% of the nodules can disappear over time [6], reflecting their reactive rather than neoplastic nature. 

Size increase and changes in imaging characteristics are the main indications for performing liver biopsies in suspicious focal hepatic lesions, especially considering that cirrhosis and deranged venous outflow may induce HCC development. However, these variations in imaging features are not specific indicators of malignancy in patients with BCS. Furthermore, to the best available knowledge, there is no evidence of malignant transformation of FNH-like RNs.

### Imaging Features

Being composed of nearly normal hepatocytes, >95% of FNH-like RNs are isodense to surrounding parenchyma on nonenhanced CT imaging (Figure 1A) [14]. In some cases, when located in the subcapsular region, they may be detected due to the alteration of liver contours. More rarely, they may appear hyperdense or hypodense compared to the liver.

After iodine contrast administration, the nodules are well evident on arterial phase images because they are hypervascularized, hence showing marked and homogeneous contrast enhancement in almost all cases (Figure 1B). Hypervascularization is atypical in cirrhotic regenerative nodules, but in FNH-like RNs associated with BCS it probably represents a compensatory response to regional loss of portal flow. On the portal and late venous phases, the FNH-like RNs usually remain isodense or slightly hyperdense (Figure 1C,D). A hypodense perinodular rim may also be observed on contrast enhanced CT imaging because of the presence of atrophic hepatic tissue around the nodule with sinusoidal congestion.

CT features of FNH-like RNs frequently change during follow-up. According to a retrospective longitudinal study by Flor et al. [15], nodules that were small (<15 mm) and homogeneously enhancing in the arterial phase became larger and showed a heterogeneous enhancing pattern, often with washout and a central scar, in 42% of cases after three years from detection.

In MRI imaging, most FNH-like RNs show a typical T1-hyperintensity (75–84%) [5,22], which is better appreciated on fat suppressed sequences. On T2-weighted images, their appearance has been variously reported [6,10,23,24], with most nodules being isointense or hypointense to the liver parenchyma and less frequently (<20%) hyperintense. Some authors suggested that these features of FN-like RNs depend on mineral deposits inside the lesions, especially copper [9,25]; however, they could also be explained by liver congestion resulting in relative T1-hypointensity and T2-hyperintensity of the perinodular parenchyma [10]. Considering that these nodules are drained by hepatic veins, the few cases of T1-hypointense or T2-hyperintense lesions may also be related to intralesional infarction when the venous outflow is compromised [14].

The presence of a central scar is another common finding in FNH-like RNs, particularly in those larger than 1 cm [10]. This scar typically appears as a hypointense, central stellate area on fat-suppressed T1-weighted imaging and a hyperintense area on T2-weighted imaging (Figure 2). Moreover, since regenerative nodules are made up of normal hepatocytes, they appear isointense or slightly hyperintense to the adjacent liver parenchyma on diffusion-weighted imaging, and there is no significant reduction in the diffusion values observed on the ADC map. These characteristics help to differentiate between regenerative and HCC nodules with reasonable confidence in patients with BCS.

As described for CT imaging, FNH-like RNs usually exhibit marked homogeneous enhancement on arterial phase fat-suppressed T1-weighted images following the administration of gadolinium-based contrast agents. In some cases, a peripheral hypointense rim may also be present due to congestion and atrophy of the surrounding liver parenchyma. This enhancement pattern differs from that of cirrhotic hyperplastic nodules, which are T2 hypointense but typically appear hypovascular after contrast administration [26]. It is worth noting that less than 5% of FNH-like RNs are hypointense or isointense in the arterial phase [6].

On portal venous phase and delayed images, FNH-like RNs usually remain slightly hyperintense or isointense compared to the surrounding parenchyma [12]. However, in almost 10% of cases on portal venous phase images and 30–40% on delayed phase images, they may exhibit washout [22]. This finding has been suggested to be a “pseudo-washout” caused by perilesional congestion, resulting in increased signal intensity of the surrounding parenchyma and relative hypointensity of the FNH-like RNs [27]. Consequently, the portal venous and delayed phases do not aid in the characterization of nodules in BCS patients. Furthermore, 18% of FNH-like RNs larger than 1 cm are reported to show hypervascularity on the arterial phase and washout on subsequent phases [16], mimicking HCC. Due to the limited specificity of these features, the Liver Imaging Reporting and Data System (LI-RADS) criteria [28] and those proposed by the American Association for the Study of Liver Diseases (AASLD) [29] and the European Association for the Study of the Liver (EASL) [30] for the diagnosis of HCC in cirrhotic patients are not suitable for patients with BCS.

On the contrary, hepatobiliary phase (HBP) images acquired after administration of hepatobiliary contrast agents can allow us to distinguish the FNH-like RNs from HCC. The benign nodules often contain ductular proliferation [7], hence appearing iso- or more often hyperintense in HBP compared to the normal parenchyma like typical FNH. For this reason, the use of hepatobiliary contrast agents on MRI is mandatory in patients with hepatic nodules associated with BCS. 

MRI features of FNH-like RNs are shown in Figure 3.

Besides changes in size, as mentioned above, the MRI features of FNH-like RNs can alter during the follow-up period (Figure 4). For example, the T1 and T2 signal intensity may change, more frequently with a shift from hyperintense to isointense on T1-weighted images and from hypointense to isointense on T2-weighted images. The enhancement pattern may vary as well, with washout acquired in 8% of cases and lost in nearly 20% of cases.

In US imaging, FNH-like RNs are often heterogeneous, but they may also appear isoechoic [31] and become challenging to detect in cases where the background liver parenchyma is altered due to congestion or fibrosis (Figure 5). Hypervascularity on Doppler flow imaging and peripheral hypoechoic rim were very common findings, but lack the specificity to distinguish regenerative nodules from malignant lesions. Contrast-enhanced ultrasound (CEUS) offers valuable support in the detection and characterization of focal liver lesions in BCS [32,33]. After intravenous administration of microbubbles, around 70% of FNH-like RNs demonstrate a rapid center-to-periphery hyperenhancing filling in the arterial phase, with one-third displaying a characteristic “spoke wheel” aspect. In the portal and delayed phases, nearly 90% of FNH-like RNs maintain consistent and homogeneous hyperenhancement, while the rest show enhancement similar to the background liver parenchyma. These features differ from those of HCC nodules, as they more commonly show washout in the portal and venous phases.

## 3. Hepatocellular Carcinoma

HCC in BCS accounts for 0.7% of all cases of primary liver malignancy [34], but the prevalence differs significantly depending on the population studied. In India and Europe, the prevalence of HCC is relatively low, at nearly 2% and 11%, respectively. In contrast, the prevalence of BCS-associated HCC is estimated to be much higher in Africa and Japan, reaching up to 40% in certain studies [35,36]. Therefore, the European Association for the Study of the Liver (EASL) recommended screening for HCC in all patients with BCS [30]. Interestingly, a large Swedish study [37] suggested that most patients with BCS have a standard risk of developing HCC, with a 5-year cumulative incidence of 0.6% and a long-term excess risk of only 2.2% compared to the general population. Conversely, in patients with cirrhosis, the cumulative incidence of HCC in BCS significantly increases, rising to approximately 3% at 5 years and up to 30% at 20 years. This highlights the pivotal role of cirrhosis as a necessary condition for developing HCC in BCS. In fact, it is assumed that chronic liver congestion and consequent fibrosis lead to an extensive centrilobular necrosis, triggering liver regenerative activity and thereby promoting hepatocarcinogenesis [38]. This model finds support in the observation that HCC is particularly frequent when BCS is caused by membranous obstruction of the inferior cava vein [39]. Furthermore, other factors such as chronic viral hepatitis may exert an additional and simultaneous influence on the development of primitive malignancies.

### Imaging Findings

HCC in BCS typically manifests as a solitary large-sized nodule (mean diameter: 5–7 cm) [40], with a heterogenous appearance in CT and MRI imaging. 

As regards CT, the sensitivity for detecting HCC > 2 cm is considered high, reaching approximately 90%, but it falls significantly when the tumor size is smaller [41]. Although dedicated studies on CT performance in detecting HCC in BCS are lacking, it is also reasonable to believe that the accuracy is lower due to the parenchymal changes caused by the venous congestion and the frequent presence of regenerative nodules. 

On MRI, the appearance of HCC is highly variable, with around 60% of lesions exhibiting hypointensity on T1-weighted images and 60% showing hyperintensity on T2-weighted images. HCC nodules usually restrict diffusion on DWI, but some lesions, particularly well-differentiated ones, may not demonstrate significant restriction compared to the background liver parenchyma. Other characteristics traditionally reported, such as nodule-in-nodule appearance, enlargement of the arterial hepatic vessels and tumor invasion of the portal system, can be observed in BCS patients as well [5,34].

After intravascular contrast administration, HCC nodules are typically hypervascular, with diffuse or, less frequently, central or peripheral, homogeneous or inhomogeneous hyperenhancement on arterial phase (APHE), followed by rapid washout in venous and delayed phase images on both CT and MRI.

However, considering that approximately 25% of HCCs do not display washout in the portal and delayed phases, while up to one-third of FNH-like RNs can show it, the specificity of this feature for the diagnosis of HCC in BCS is relatively low. As reported by Van Wettere et al. [22], the association of homogeneous APHE and homogeneous washout was identified in one-third of benign lesions and in half of HCC cases. The resulting sensitivity and specificity for diagnosing HCC were 50% and 70%, respectively. When considering the combination of any type of APHE (homogeneous, peripheral or central) and any type of washout (homogeneous or peripheral), a sensitivity of 100% and a specificity of 61% were achieved. For this reason, given the risk of an unacceptable rate of false-positive results, the LI-RADS and AASLD/EASL criteria for the non-invasive diagnosis of HCC cannot be applied in BCS patients [28,29,30]. The differential diagnosis between FNH-like RNs and HCC becomes even more challenging due to the potential for the former to increase in size and/or in number. Therefore, it is important to consider additional features supportive of diagnosis of HCC, such as T1 hypointensity, hyperintensity in T2-weighted and high b-value DWI, and absence of a central scar on MRI [42], as shown in Figure 6. 

Hepatobiliary contrast agents provide valuable help in the differential diagnosis, as HCC most often demonstrates a hypointense signal in HBP. In the above-mentioned study [22], the combination of this feature with APHE and washout proved effective in distinguishing all HCC lesions from FNH-like RNs. Indeed, benign regenerative lesions typically appear isointense-to-hyperintense compared to the normal liver in HBP, regardless of the presence of washout on portal venous or delayed phase images. 

In US imaging, HCC nodules are frequently heterogeneous in appearance, but they can display various aspects, making it impossible to differentiate them from FNH-like RNs based solely on echogenicity. Hypervascularity on Doppler flow imaging and the presence of peripheral hypoechoic rim are also common findings, which do not provide significant assistance in the differential diagnosis. In CEUS imaging, the majority of HCC nodules demonstrate heterogeneous or homogeneous hyperenhancement during the arterial phase, which is consistent with the behavior observed in other imaging modalities. Additionally, in the study conducted by Zhang et al. [32], center-to-periphery enhancement and the “spoke wheel” pattern were never observed in HCC nodules. Venous washout remains a typical feature, with around 80% of the malignant lesions exhibiting a hypoechoic appearance compared to the liver parenchyma in the portal phase, and nearly all of them demonstrating it in the delayed phase. In these terms, CEUS may help to differentiate benign regenerative nodules from HCC in BCS patients.

Table 1 summarizes the imaging features of HCC and FNH-like RNs in CT, MRI, and US.

Complementing the imaging findings with measurement of alpha-fetoprotein (AFP) and evidence of rapid growth can increase specificity and is relevant for diagnostic and monitoring purposes. In presence of HCC, AFP levels are usually positively correlated with tumor size [43] and tend to increase at successive determinations. A cut-off of 15 ng/mL is recommended to define an elevated serum AFP level, which demonstrates excellent positive (up to 100%) and negative (90%) predictive values for the diagnosis of HCC [44]. Moreover, the sensitivity and specificity of the combination of a hypointense signal in HBP and AFP > 15 ng/mL are up to 92% and 98%, respectively, for the diagnosis of HCC [22]. When hepatic lesions in BCS show characteristic features of FNH-like RNs and demonstrate low AFP levels, patients can undergo a routine follow-up evaluation at six months through clinical, laboratory, and imaging assessments. Moucari et al. [44] advocated an alternative strategy to improve the sensitivity for detecting malignant lesions, which involves a 3-monthly surveillance during the initial year following the identification of a liver nodule. Subsequently, if no changes are observed, the follow-up intervals can be extended to 6 months. In case of atypical imaging findings, an elevation in AFP levels, an increase in size on two successive imaging examinations or other significant changes during the follow-up period, a liver biopsy should be recommended, especially when the number of lesions is ≤3 and/or the nodule diameter is ≥3 cm [12,27]. In this regard, it is important to note that BCS patients are at heightened risk of bleeding due to anticoagulation therapy, whereas its discontinuation may result in recurrent thrombosis.

CT or US are commonly used to guide biopsies, helping to identify the correct target when multiple nodules are present and ensuring the representative sampling of the lesion. Real-time monitoring during the procedure further enhances accuracy and safety, reducing the risk of complications.

Imaging also supports therapeutic interventions for suspicious nodules. For instance, by detecting hepatic venous obstruction it enables the optimal timing of liver resection, ensuring that the procedure is scheduled when venous outflow is restored. This approach aims to reduce liver congestion and minimize complications after surgery [45]. Additionally, imaging can detect the signs of portal hypertension, which may be present even in the absence of cirrhosis. In these cases, a surgical resection may be performed after controlling portal pressure, such as through TIPS placement, or alternative therapeutic options can be considered [46].

## 4. Other Focal Liver Lesions

Hemangioma, true focal nodular hyperplasia, adenoma, dysplastic nodule, and necrotic regenerative nodule are other focal liver lesions that may coexist in patients with BCS (Figure 7). 

In general, they present the same radiological characteristics as in the general population; therefore, an exhaustive discussion is deferred to other sources [18,42]. It is worth mentioning that the radiological and histological features of these lesions exhibit variable overlap, so the differential diagnosis often relies on clinical information and monitoring. For instance, the necrotic regenerative nodule is hyperintense on T2-weighted and hypointense in T1-weighted images, shows diffusion restriction in DWI and demonstrates hypointensity in HBP, hence mimicking HCC [12]. The absence of enhancement throughout the dynamic sequences helps with orienting the diagnosis, although 10–20% of HCC nodules are not hypervascular. Another notable example is focal nodular hyperplasia, which is virtually indistinguishable from FNH-like RNs, yet it conventionally develops in healthy parenchyma and tends to present as a solitary lesion.

Hypervascular metastases, including those from breast cancer, renal cell carcinomas, neuroendocrine tumors, and melanomas [47], should also be considered in the differential diagnosis of HCC. Anecdotally, rare liver tumors like cholangiocarcinoma and epithelioid hemangioendothelioma have been reported to occur in the background of primary BCS [48,49]. 

Finally, in BCS is possible to encounter pseudo-lesions. A typical example is the selective hypertrophy of the caudate segment (or any other segments with preserved venous drainage) in chronic BCS, which may take a tumor-like appearance (Figure 8) [50]. The hypertrophied segments may be hyperdense on unenhanced CT scan or exhibit T1-hyperintensity and mild T2-hypointensity in MRI with elevated restriction in DWI due to increased cellularity. The hypertrophy-induced pseudo-lesions usually display the regular enhancement pattern observed in normal liver, but this can be misleading in cases of extensive perfusion alterations due to venous congestion or in advanced stages of the disease. A useful clue for the differential diagnosis is the identification of normal, non-distorted intrahepatic vessels within the hypertrophied segment. Perfusion-associated pseudo-lesions can also occur, especially in acute or subacute BCS with proximal venous obstruction [51]. After contrast agent administration, areas of venous congestion may show a peculiar “mosaic pattern” (Figure 9), with patchy and/or peripheral areas of transient enhancement, instead of the classical wedge-shaped alterations [52]. These perfusion anomalies may simulate the presence of lesions, especially considering that the areas of parenchymal congestion appear heterogeneously hyperintense on T2-weighted and hypointense on T1-weighted sequences compared to the caudate, where the drainage is usually preserved (Figure 10).

## 5. Conclusions

In patients with BCS, new onset focal liver lesions are commonly detected, with benign regenerative nodules being the most frequently encountered. This population has a higher predisposition to hepatocarcinogenesis, provided that chronic congestion has already induced cirrhosis. Arterial hyperenhancement and contrast washout in portal venous and delayed phases alone are not specific enough to differentiate between FNH-like RNs and HCC nodules. Integration of information from HBP acquisitions, serum AFP levels, and CEUS provides valuable assistance in the differential diagnosis. In general, CT, MRI and US are crucial for the tempestive detection and accurate classification of focal liver lesions in BCS, facilitating optimal patient management.

## Figures and Tables

**Figure 1 diagnostics-13-02346-f001:**
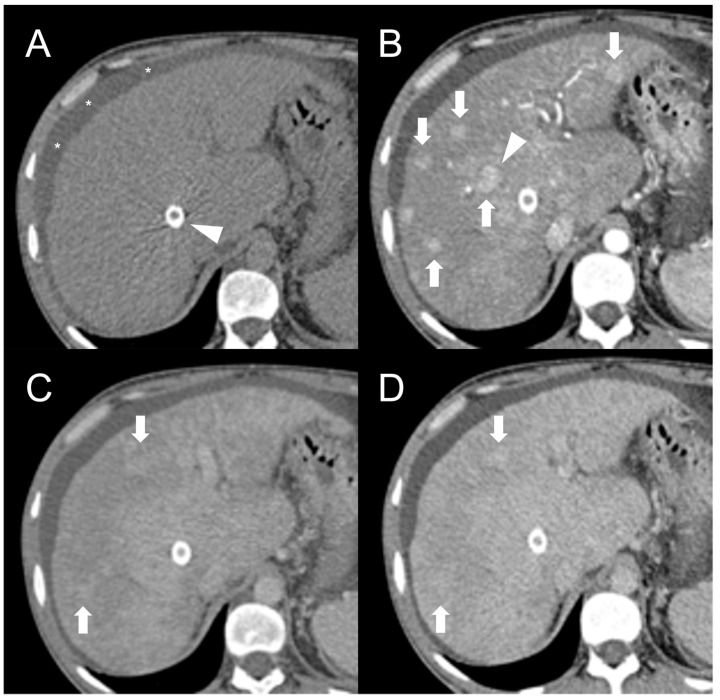
CT examination of 36-years-old male patient with Budd–Chiari syndrome. On unenhanced images (**A**), the liver parenchyma appears relatively homogeneous, and the presence of TIPS stent (arrowhead) and perihepatic ascites (asterisks) can be observed. On arterial phase (**B**), multiple homogeneously enhancing FNH-like RNs can be visualized in the liver (arrows), some exhibiting a hypodense perinodular rim (arrowhead) due to atrophic hepatic tissue with congested sinusoids. In both portal venous (**C**) and delayed phase (**D**), the regenerative nodules (arrows) become iso-dense or slightly hyperdense compared to the surrounding inhomogeneous liver parenchyma, making them difficult to detect.

**Figure 2 diagnostics-13-02346-f002:**
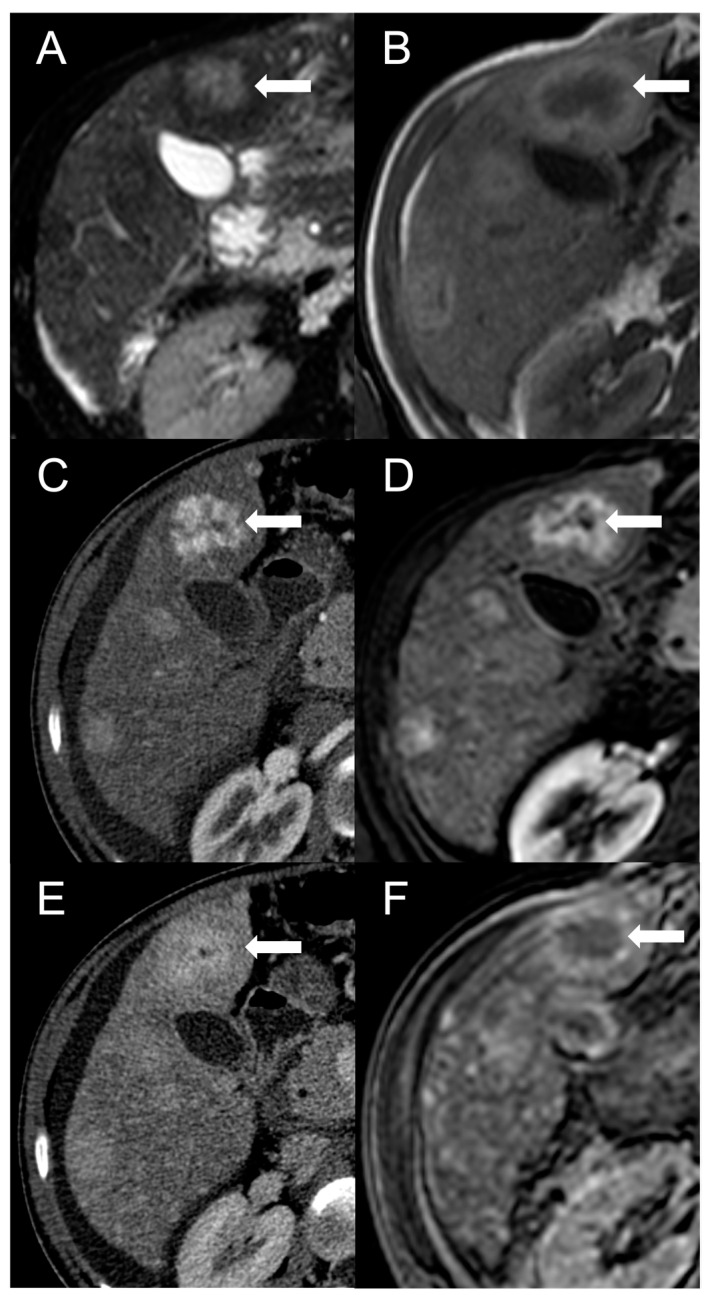
FNH-like RN with central scar (arrow) in a 34-year-old male patient with Budd–Chiari syndrome. In MRI, the scar is typically hyperintense on fat-saturated T2-weighted images (**A**) and hypointense in T1-weighted images (**B**). In the arterial phase after contrast agent administration, the scar can be identified as a central stellate area that appears hypodense in CT (**C**) and hypointense in MRI (**D**). In the CT delayed phase, the nodule shows increased density, but a central hypodense component representing the scar remains visible (**E**). In gradient-echo T1-weighted images acquired in the hepatobiliary phase (**F**), the scar is clearly depicted as a central low-signal area.

**Figure 3 diagnostics-13-02346-f003:**
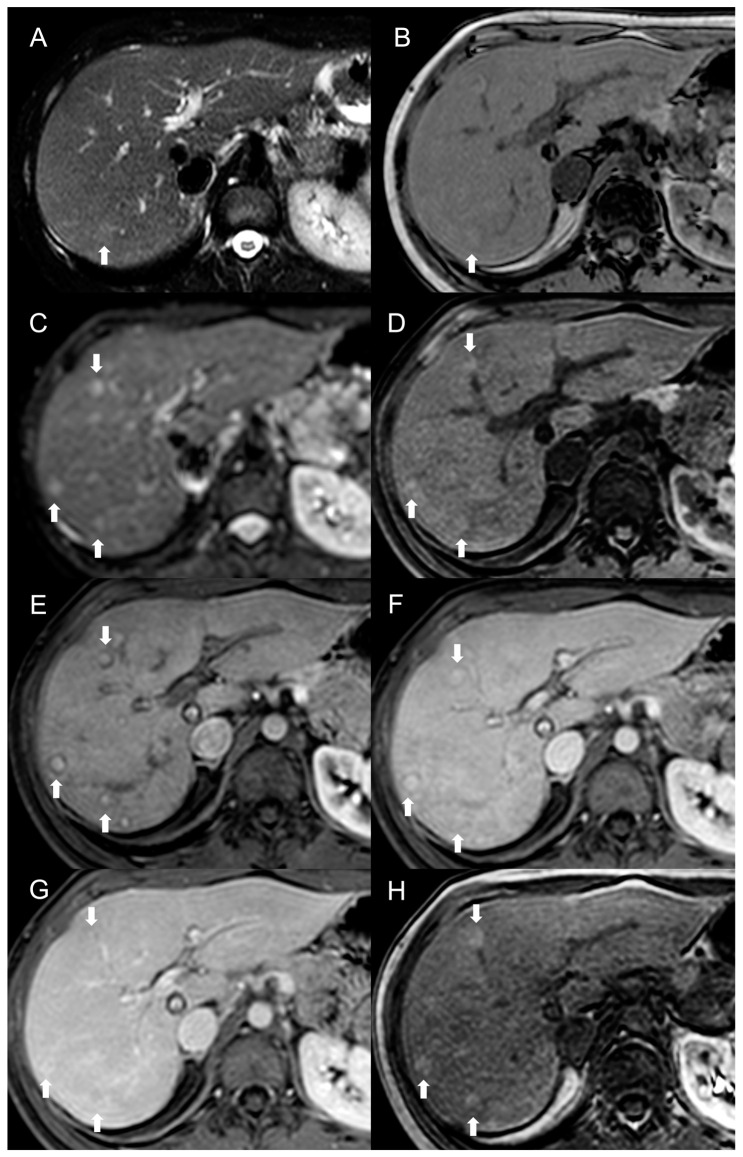
MRI examination of a 29-year-old female with Budd–Chiari syndrome. On fat-saturated T2-weighted images (**A**) FNH-like RNs are barely discernable as they appear isointense to the liver parenchyma; occasionally, areas of very slightly hyperintense signal (arrow) can be identified, possibly related to congested nodules or central scar. On out-of-phase T1-weighted images (**B**) the presence of nodular lesions is more appreciable, as FNH-like RNs (arrow) appear slightly hyperintense compared to the surrounding liver parenchyma. On DWI (b-value = 600) (**C**), FNH-like RNs may show a slight heterogeneous hyperintensity due to central scar or congestion. On unenhanced gradient-echo T1-weighted images (**D**), the nodules (arrows) appear hyperintense, while on arterial phase (**E**) they show vivid enhancement with a peripheral hypointense rim due to hepatic tissue congestion. On portal venous (**F**) and delayed (**G**) phases, the nodules become progressively isointense to the liver. On the T1-weighted images (flip angle = 30°) in the hepatobiliary phase (**H**), FNH-like RNs show hyperintensity compared to the surrounding liver parenchyma.

**Figure 4 diagnostics-13-02346-f004:**
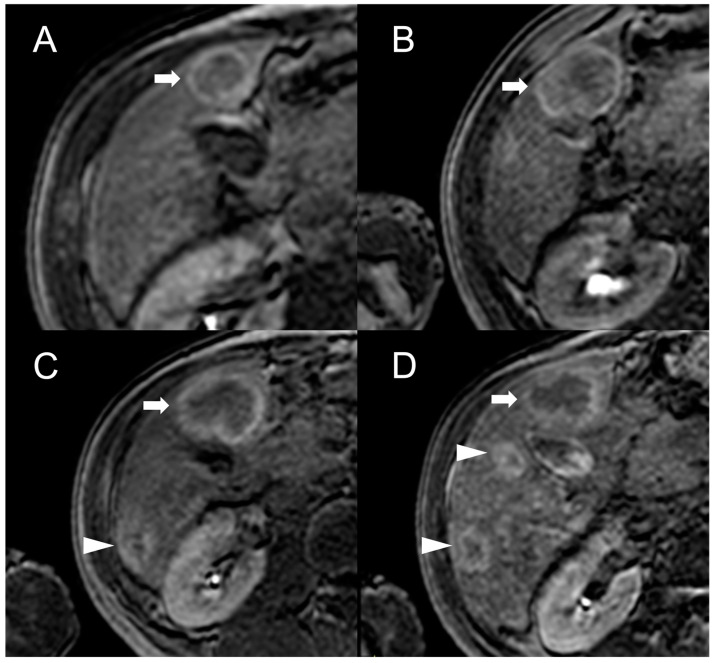
MRI examinations of a 31-year-old male patient with Budd-Chiari syndrome performed in 2016 (**A**), 2017 (**B**), 2018 (**C**), and 2020 (**D**). Gradient-echo T1-weighted sequences (flip angle = 30°) acquired in the hepatobiliary phase revealed a 25 mm FNH-like RN (arrows) with central scar and peripheral hyperintensity in 2016 (**A**). The lesion size grew to 41 mm after 1 year (**B**) and further to 44 mm after 2 years (**C**), but it reduced to 35 mm in 2020 (**D**). Additionally, new FNH-like RNs appeared over time (arrowheads in **C**,**D**).

**Figure 5 diagnostics-13-02346-f005:**
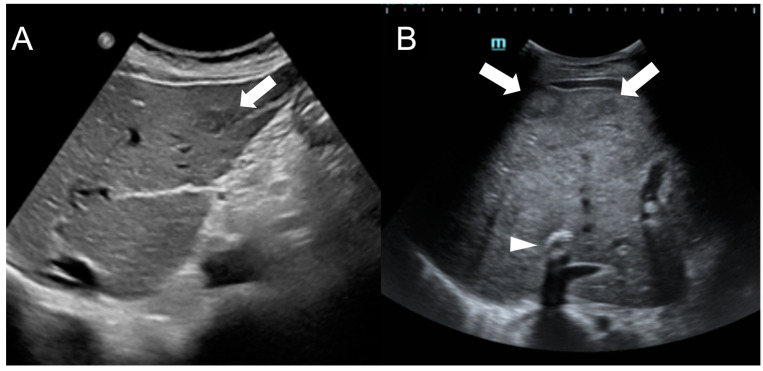
Ultrasound images of a 59-year-old female patient (**A**) and a 34-year-old male patient (**B**) with Budd-Chiari syndrome. In both cases, FNH-like RNs appear as hypoechoic lesions (arrows) within the heterogeneous liver parenchyma, occasionally exhibiting a central hyperechoic scar. A TIPS stent is visible in case B (arrowhead).

**Figure 6 diagnostics-13-02346-f006:**
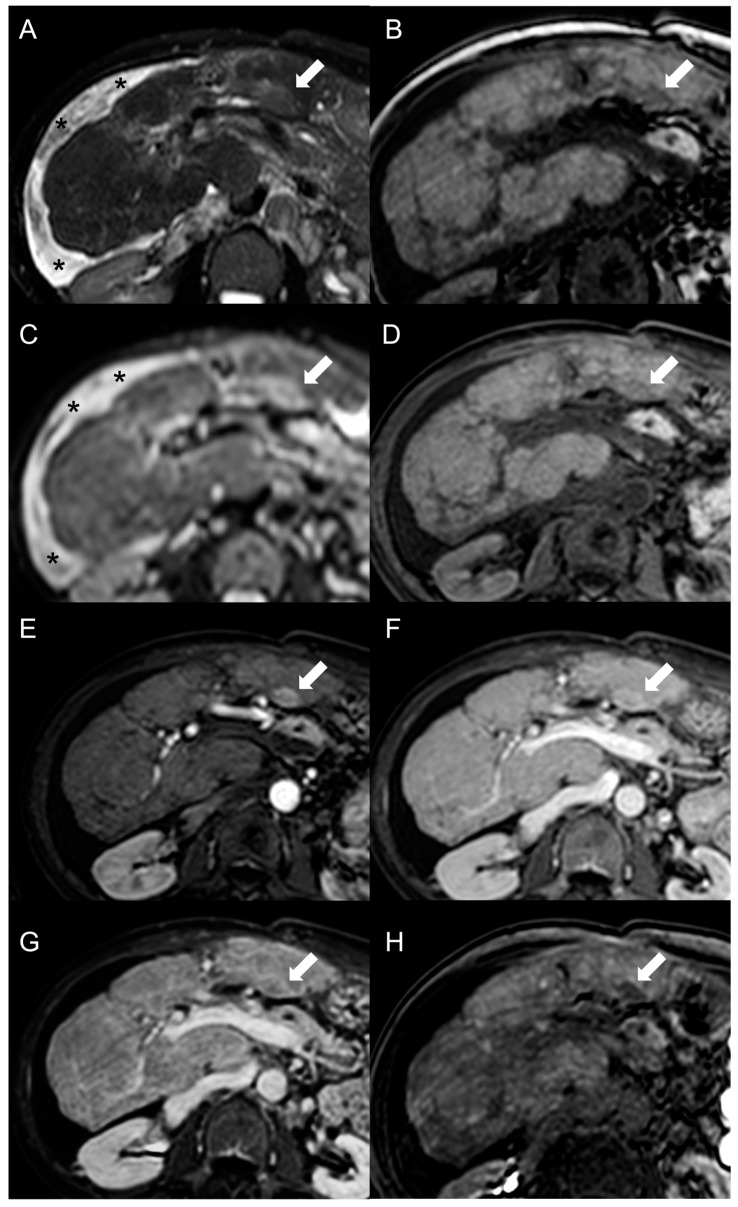
MRI examination of a 67-year-old female patient with Budd–Chiari syndrome reveals a new focal lesion of 13 mm in the left lobe (arrow). The lesion appears hyperintense on T2-weighted images (**A**), hypointense on T1-weighted images (**B**,**D**) and shows restricted diffusion in high-b-value DWI (**C**). After contrast agent administration, the lesion demonstrates marked enhancement on T1-weighted arterial phase images (**E**), followed by central washout with enhancing capsule on T1-weighted portal venous (**F**) and delayed (**G**) phases. On T1-weighted images in the hepatobiliary phase (**H**) the lesion shows a hypointense signal, suggesting a diagnosis of HCC. Perihepatic ascites (asterisks) can be observed on T2-weighted images (**A**) and DWI (**C**).

**Figure 7 diagnostics-13-02346-f007:**
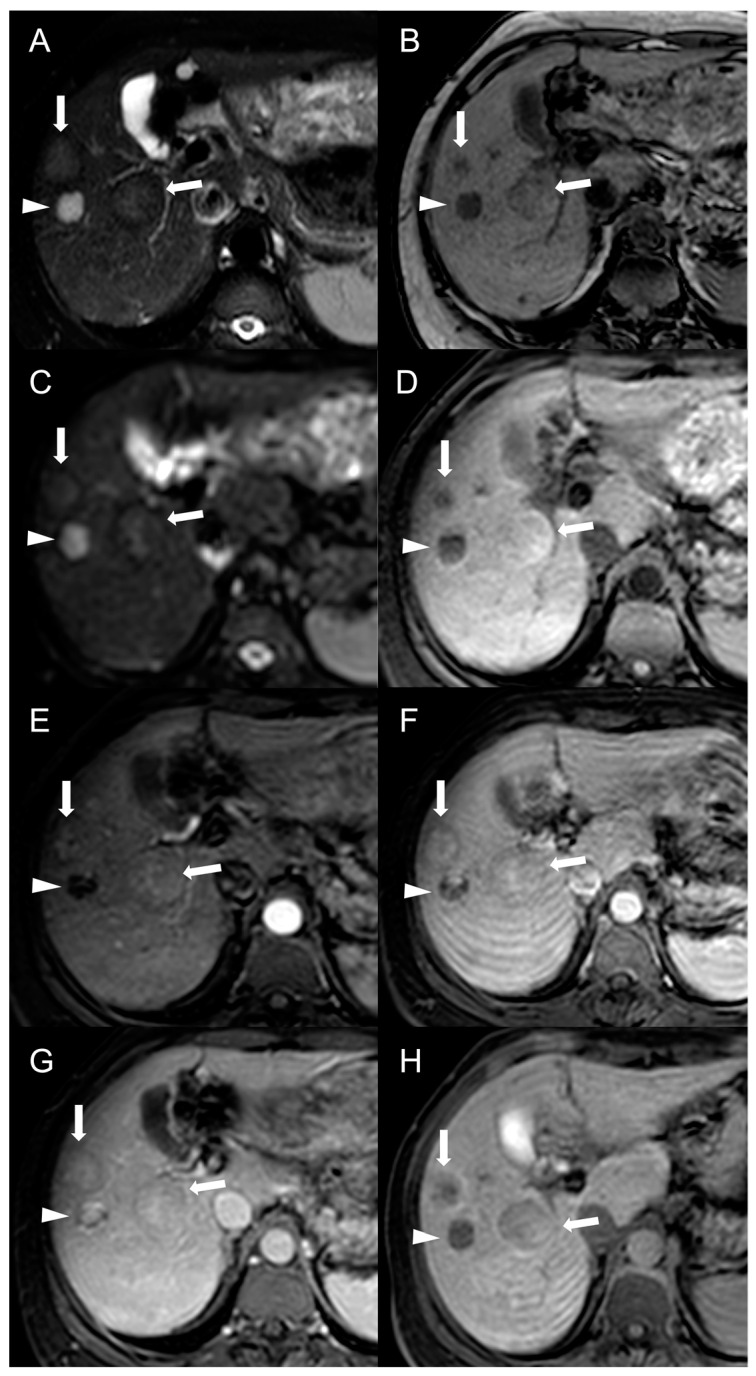
MRI examination of a 49-year-old female patient with Budd–Chiari syndrome reveals multiple lesions, including two large FNH-like RNs (arrows) in V-VI liver segment. On fat-saturated T2-weighted images (**A**), these nodules are iso-hypointense with an inhomogeneous central hyperintensity and a low-signal perinodular rim. On T1-weighted images (**B**) the FNH-like RNS appear hyperintense, one of them exhibiting a central hypointensity corresponding to the central scar. In DWI (b-value = 600) (**C**), they show the same signal intensity as the adjacent liver parenchyma, with appreciable low-signal peripheral rim. On T1-weighted unenhanced (**D**), arterial (**E**), portal (**F**), and delayed (**G**) phase images, the FNH-like RNs demonstrate inhomogeneous arterial enhancement and isointense signal on the portal and delayed phases; the hypointense perinodular rim, representing congested hepatic tissue, remains noticeable. In the hepatobiliary phase (**H**), the FNH-RNs display inhomogeneous iso-hyperintense signals compared to the surrounding liver parenchyma. A third lesion, compatible with a hemangioma (arrowhead), is also visible, showing homogeneous hyperintensity on fat-saturated T2-weighted images (**A**) and DWI (b-value = 600) (**C**), hypointensity on T1-weighted images (**B**), slow centripetal globular enhancement on arterial (**E**), portal (**F**), and delayed phase (**G**) images, and hypointense signals in the hepatobiliary phase (**H**).

**Figure 8 diagnostics-13-02346-f008:**
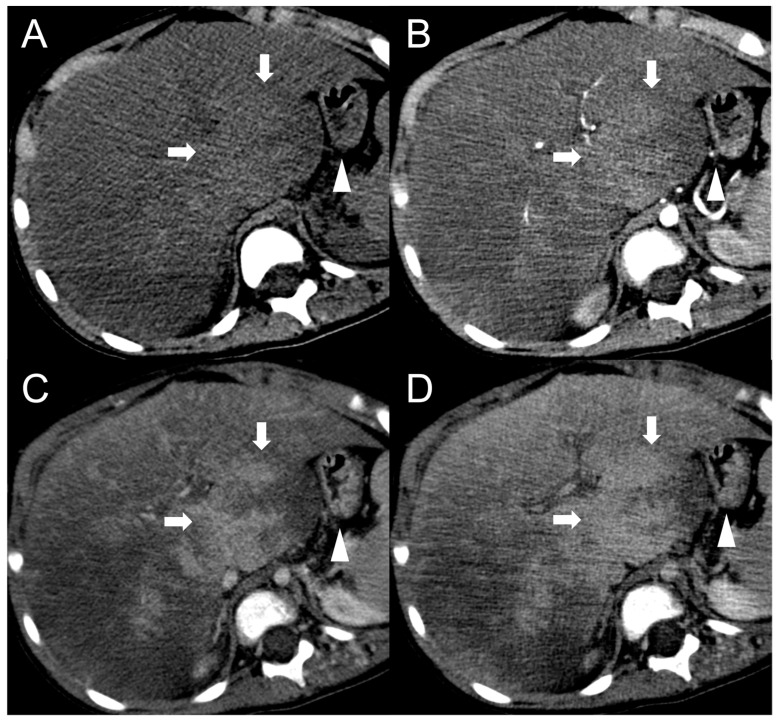
CT examination of a 5-year-old female patient with Budd–Chiari syndrome revealing a pseudo-lesion caused by hypertrophy of the caudate lobe. The caudate lobe (arrows) shows slight hyperdensity on unenhanced images (**A**), with heterogeneous contrast enhancement in the arterial (**B**), portal (**C**) and delayed phase (**D**). Notably, this represents a normal enhancement pattern due to preserved venous drainage, in contrast to the surrounding congested liver parenchyma. The hypertrophied caudate lobe also causes compression on the stomach (arrowheads), contributing to its mass-like appearance.

**Figure 9 diagnostics-13-02346-f009:**
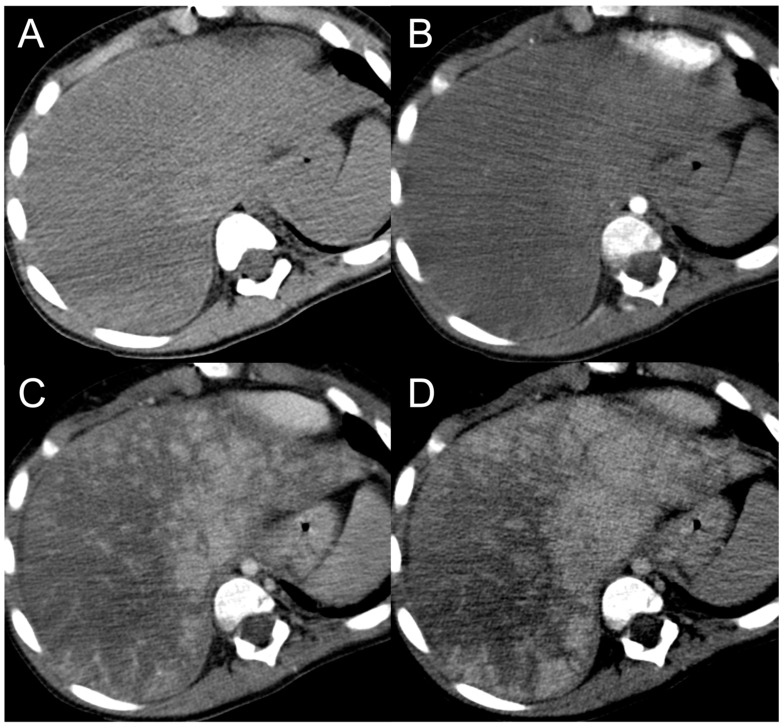
CT examination of a 5-year-old female patient with Budd–Chiari syndrome. Unenhanced (**A**) and arterial phase (**B**) images show diffuse inhomogeneity of the liver parenchyma. In the portal phase (**C**) a patchy “mosaic pattern” enhancement is observed, with ill-defined nodular pseudo-lesions. In the delayed phase (**D**), these lesions tend to merge and disappear, indicating their perfusion-related nature.

**Figure 10 diagnostics-13-02346-f010:**
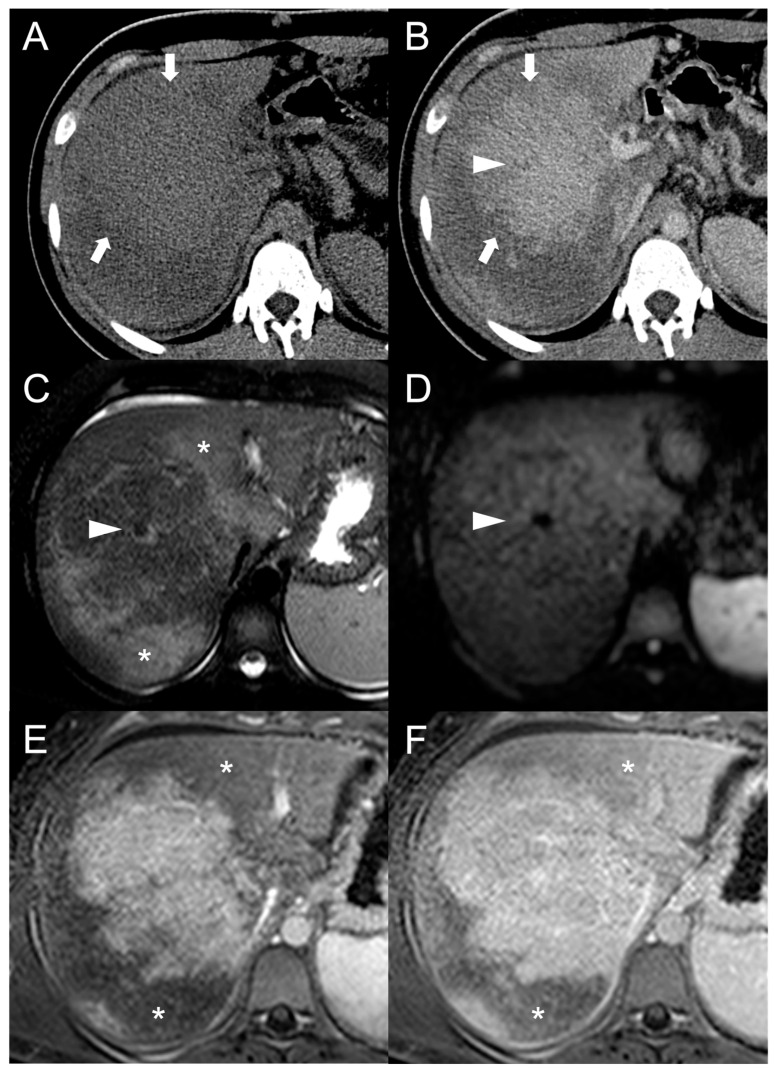
Perfusion-related pseudo-lesion in a 19-year-old male patient with acute Budd–Chiari syndrome. On the unenhanced (**A**) and portal venous phase (**B**) CT images a large mass-like alteration with irregular and ill-defined margins is appreciable in the central region of the right liver lobe (arrows). The thrombosed right hepatic vein (arrowhead) is also visible. On MRI, the fat-saturated T2-weighted imaging (**C**) demonstrates that the pseudo-lesion consists of normal liver tissue surrounded by congested hyperintense parenchyma (asterisks). On DWI (**D**), no areas of abnormal restriction are observed as well. Gradient-echo T1 weighted images in the portal (**E**) and hepatobiliary (**F**) phases confirm the perfusion-related nature of the pseudo-lesion, showing delayed and decreased enhancement in the congested areas (asterisks).

**Table 1 diagnostics-13-02346-t001:** Imaging findings of FNH-like regenerative nodules and hepatocellular carcinoma in Budd–Chiari syndrome.

	FNH-like RN	HCC
CT findings		
Unenhanced scan	Isodense	Heterogenous (commonly iso-hypodense)
Arterial phase	Hyperdense (homogenous APHE)	Hyperdense (non-rim APHE)
PVP/DP phase	Isodense/slightly hyperdense	Hypodense (early washout—75%)
Hypodense perinodular rim	+/−	−
Central scar	+/−	−
MRI findings		
T1	Hyperintense or Isointense (75–84%)	Heterogenous (60% hypointense)
T2	Hypointense or isointense (>80%)	Heterogenous (60% hyperintense)
DWI	−/(+)	+
Arterial phase	Hyperintense (homogeneous APHE)	Hyperintense (non-rim APHE)
PVP/DP	Hyperintense or isointense (30–40% hypointense)	Hypointense (early washout—75%)
HBP	Hyperintense or isointense	Hypointense
Central scar	+	−
Hypointense perinodular rim	+	−
Enhancing capsule	−	+
Ultrasound findings		
Echogenicity	Heterogenous or isoechoic	Heterogenous
Doppler flow imaging	(−)/+	(−)/+
Hypoechoic peripheral rim	−/+	−/+
CEUS		
Arterial phase	Rapid center-to-periphery filling (30% spoke wheel pattern)	Hyperechoic
PVP/DP	Hyperechoic (90%) or isoechoic	Hypoechoic (early washout—80–100%)

FNH-like RN: nodular hyperplasia-like rigenerative nodule; HCC: hepatocellular carcinoma; APHE: arterial phase hyperenhancement; PVP/DP: portal venous phase/delayed phase; HBP: hepatobiliary phase; CEUS: contras agent ultrasound.

## Data Availability

Not applicable.

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
