# Peer review of "Focal Liver Lesions in Budd-Chiari Syndrome: Spectrum of Imaging Findings"

_diagnostics, 2023, doi:10.3390/diagnostics13142346_

Round 1

Reviewer 1 Report

This is a well-written review with sufficient images to support the arguments made. 

Minor suggestion

In "Conclusion" part some sentences need to be rewritten for better presentation.

The manuscript is well organized and english structure also good. The second sentence in conclusion part can be rewritten. Otherwise the manuscript is fine to be considered for publication. Technically it looks sound.

English quality is good.

Author Response

Dear Reviewer,

thank you for taking time to review our work, we appreciate your comments and support. As you suggested, we improved the Conclusion section of our manuscript for better clarity and readability. 

Reviewer 2 Report

Rizzetto F et al's manuscript reviews focal liver lesions in Budd-Chiari syndrome. Specially they explained the spectrum of imaging findings in BCS.

Ratio of BCS patients with HCC is too much in Africa and Japan. I could learn the pathology and diagnosis of BCS.  

Author Response

Dear Reviewer,

thank you for taking time to review our manuscript and your useful feedback. Regarding the epidemiological data we presented, we relied on the extensive review by Ren (see 10.1097/MEG.0b013e32835eb8d4), which reported ratios as high as 40-50% for Japan and Africa. This is probably due to to the fact that comprehensive and recent epidemiological works on this topic are not available. In light of this, we have revised and mitigated the related sentence in our manuscript.

Reviewer 3 Report

After delivering the necessary general information on Budd-Chiari syndrome the authors describe the non invasive diagnostic modalities for the detection and characterization of focal liver lesions in BCS.

The review is clear, comprehensive and of relevance to the field and has the merit of containing all the information available at the moment.

I found no review covering this specific topic.

The information covers the topic.

The presentation is clear, comprehensive and well documented.

The references are appropriate, up-to-date and contain 49 titles.

I found no self-citations.

The figures (10) are good and illustrate the text . Figures 8,9,10 should precede the conclusions.

The table gives an overview on the topic.

The conclusions are coherent and connected to the content.

In my opinion the paper fits the journal and the language is correct and understandable.

 I recommend the paper to be accepted.

Author Response

Dear Reviewer,

thank you very much to take time to revise our manuscript, we are delighted for your positive feedback.